# Antibiotic persistence of intracellular *Brucella abortus*

**Selma Mode**¤☉, **Maren Ketterer** ☉, **Maxime Québatte** *☉, **Christoph Dehio** *

Biozentrum, University of Basel, Basel, Switzerland

☉ These authors contributed equally to this work.
¤ Current address: F. Hoffmann-La Roche Ltd, Product Development Global Clinical Operations (PDG), Basel, Switzerland
* m.quebatte@unibas.ch (MQ); christoph.dehio@unibas.ch (CD)

**Data Availability Statement:** The data underlying all figures presented in this study have been included as Supplementary material (S1 Data).

**Funding:** This work was supported by the Swiss National Science Foundation grant

## Abstract

### Background

Human brucellosis caused by the facultative intracellular pathogen *Brucella* spp. is an endemic bacterial zoonosis manifesting as acute or chronic infections with high morbidity. Treatment typically involves a combination therapy of two antibiotics for several weeks to months, but despite this harsh treatment relapses occur at a rate of 5–15%. Although poor compliance and reinfection may account for a fraction of the observed relapse cases, it is apparent that the properties of the infectious agent itself may play a decisive role in this phenomenon.

### Methodology/Principal findings

We used *B. abortus* carrying a dual reporter in a macrophage infection model to gain a better understanding of the efficacy of recommended therapies *in cellulo*. For this we used auto-mated fluorescent microscopy as a prime read-out and developed specific CellProfiler pipelines to score infected macrophages at the population and the single cell level. Combining microscopy of constitutive and induced reporters with classical CFU determination, we quantified the protective nature of the *Brucella* intracellular lifestyle to various antibiotics and the ability of *B. abortus* to persist *in cellulo* despite harsh antibiotic treatments.

### Conclusion/Significance

We demonstrate that treatment of infected macrophages with antibiotics at recommended concentrations fails to fully prevent growth and persistence of *B. abortus in cellulo*, which may be explained by a protective nature of the intracellular niche(s). Moreover, we show the presence of *bona fide* intracellular persisters upon antibiotic treatment, which are metabolically active and retain the full infectious potential, therefore constituting a plausible reservoir for reinfection and relapse. In conclusion, our results highlight the need to extend the spectrum of models to test new antimicrobial therapies for brucellosis to better reflect the *in vivo* infection environment, and to develop therapeutic approaches targeting the persister subpopulation.

310030B_201273 and National Centre of Competence in Research AntiResist grant 180541 (both to C. D.). The funders had no role in study design, data collection and analysis, decision to publish, or preparation of the manuscript.

**Competing interests:** The authors have declared that no competing interests exist.

## Author summary

Brucellosis is a zoonosis endemic to many low- and middle-income countries around the world. Therapies recommended by the WHO are comprised of at least two antibiotics for several weeks, sometimes months. Relapses are frequent despite these harsh treatments. The underlying reasons for these relapses, besides reinfection and non-compliance to treatment, are unknown. Our study shows that *Brucella abortus* can form so called "persisters" in rich broth but also inside macrophages. This small bacterial subpopulation survives antibiotic treatment and resumes growth after removal of the antibiotics and could therefore serve as a reservoir for relapses in human brucellosis. Furthermore, we show that the intracellular lifestyle of *Brucella* has protective properties against recommended antibiotics as observed for other intracellular pathogens, highlighting the necessity to develop new infection models to assess antibiotic efficacy.

## Introduction

Brucellosis, caused mostly by *Brucella melitensis*, *Brucella abortus* (*B. abortus*) and *Brucella suis*, is the most common zoonotic bacterial disease worldwide with about half a million yearly cases reported from 91 countries, ranging from South to Central America, Africa to Asia, and the Mediterranean region to the Arabian Peninsula [1–4]. The number of unreported cases is estimated to be 10 to 25 times higher due to the unspecific nature of the symptoms experienced during acute brucellosis, such as fever, malaise, and arthralgia [5]. Human brucellosis can culminate in a severe chronic disease with symptoms lasting more than 12 months, accompanied by different complications such as endocarditis, arthritis, and meningoencephalitis [6–9]. It is a debilitating disease that is rarely fatal, but has a high disease burden, or as the TIME magazine reported in 1943: "...*the disease rarely kills anybody. But it often makes a patient wish he were dead*" [10].

In livestock, brucellosis leads to reduced fertility and a significant decline in milk production, resulting in very high economic losses [11–16]. Because of the high impact of brucellosis, the WHO classified the disease as one of seven "*neglected endemic zoonotic diseases of particular interest*" [17, 18]. Currently there is no licensed vaccine for humans available. Moreover, the current livestock vaccinations with live attenuated strains are known to cause disease in humans and occasionally abortions in vaccinated livestock and are therefore not a suitable solution for large scale eradication of the disease (reviewed in [4, 19, 20]). Human infection occurs most commonly via consumption of unpasteurized dairy products and undercooked meat but is also an important occupational disease for farm and abattoir workers, as well as veterinarians and laboratory personnel, where infection occurs through inhalation, conjunctival or skin contamination [4, 21–23]. The pathogen enters regional lymph nodes through the mucosal membranes of the respiratory and digestive tract [24], and is disseminated into organs of the reticuloendothelial system such as spleen, liver, lungs, and bone marrow (reviewed in [22, 25]). At the cellular level, *Brucella* is able to invade and replicate inside host cells [26], such as macrophages [27], dendritic cells [28], and epithelial cells [29]. Intracellular survival is achieved by the restricted fusion of the *Brucella*-containing vacuole (BCV) with lysosomes and results in the formation of a microcolony in a compartment continuous with the endoplasmic reticulum (ER) [26, 30]. Egress from the infected cells to start a new infection cycle seems to be initiated via interaction with autophagic host proteins [31–34]. These processes are facilitated

by the activity of the VirB family type 4A secretion system, utilized to deliver effector proteins into the host cell to orchestrate cellular functions in the pathogen's favor [35–45].

The current regimen to treat human brucellosis is still based on the 1986 recommendations of the WHO and consists in either a combination of doxycycline for 45 days with intramuscular streptomycin for 2–3 weeks or doxycycline plus rifampicin for 6 weeks [46, 47]. Substitution of streptomycin by gentamicin was later proposed [46, 47] and seems to be at least as effective [46–49] but also requires parenteral administration. Importantly, relapse rates or clinical failures remain high despite these harsh therapies, with an estimation varying between 5–15% [21, 50, 51]. Relapses usually occur within 6 months after completion of treatment and are typically not due to the emergence of stable antibiotic resistance [52, 53], although resistance against rifampicin has been observed [54–57]. Lack of compliance to the long duration of the treatments and reinfection(s) are possible reasons for the clinical failure [58] but in many cases the underlying causes of relapses remain elusive [59, 60]. Triple therapy regimens are used for complicated brucellosis cases associated with arthritis, spondylitis, and endocarditis and consist of streptomycin or gentamicin plus doxycycline and rifampicin for at least 8 weeks. Although these triple regimens have better efficacy and less treatment failure than dual therapies, they remain costly and come with more side effects and are therefore only used for complicated cases of Brucellosis (reviewed in [60]).

It is patent that *Brucella*'s ability to invade and replicate intracellularly constitutes a hurdle for the successful treatment of Brucellosis, as for its ability to disperse in and colonize a broad variety of niches within its host. Moreover, the possibility that *Brucella* exhibits a "persister" phenotype during infection could, as recently suggested *in vitro* [61, 62], constitute an additional complexity layer towards successful therapy. Indeed, so-called persisters constitute a challenge for clinical microbiology due to their multidrug-tolerant manifestation [63–70]. Persisters are defined as a bacterial subpopulation that transiently displays an antibiotic tolerance phenotype, without the need for a genetic modification [71–73]. A hallmark of persisters is their ability to regenerate a normal bacterial population upon removal of the threat, given that the proper nutrients are available. Mechanisms underlying persister formation are still a matter of debate and are likely diverse and possibly differ between pathogens. It is however generally accepted that any stress condition that reduces bacterial growth will result in an increase of bacterial persisters [74–76], conditions are eventually encountered by pathogenic bacteria during host infection.

In this study, we sought to establish an *in cellulo* infection model for *Brucella* to assess the efficacy of antimicrobial treatments and possibly test new strategies to eradicate the pathogen. We used *B. abortus* carrying a dual color reporter for constitutive vs. induced gene expression, the murine macrophage cell line RAW264.7, and automated fluorescence microscopy as a prime read-out. We developed specific CellProfiler pipelines to score our infection model, both at the population and at the single cell level and used these tools to evaluate the effect of the recommended therapies for human Brucellosis. Combining microscopy of constitutive and induced reporters with classical CFU determination, we quantified the protective nature of *Brucella's* intracellular niche(s) to various antibiotics. Finally, we demonstrated the presence of *bona fide* intracellular persisters, and showed that these surviving, metabolically active bacteria retained their full infectious potential, thus constituting a plausible reservoir for reinfection and relapse.

## Methods

### Bacterial cultures

All manipulations with live *Brucellae* were performed according to standard operational procedures (SOPs) in a biosafety level 3 (BSL-3) facility at the Swiss Tropical and Public Health

Institute in Basel, Switzerland. Strains used in this study include *B. abortus* 2308 wild-type (*B. abortus* WT), *B. abortus* 2308 Δ*virB9* (*B. abortus* Δ*virB9*) [77] and *B. abortus* 2308 carrying the plasmid pAC042.8 (*apht::dsRed*, *tetO::tetR-GFP*, S1A Fig) (*B. abortus*) [78], which was used to differentiate between transcriptionally active and inactive bacteria. The strain carrying plasmid pAC042.8 constitutively expresses dsRed and expresses GFP from a tetracycline-inducible promoter, which was validated by induction with 100 ng/ml anhydrotetracycline (aTc, Sigma-Aldrich 97919) (S1A–S1C Fig). *Brucella* strains were stored as frozen aliquots in 10% skim milk at -80˚C. Bacteria were grown at 37˚C with agitation (100 rpm) in tryptic soy broth (TSB, Sigma-Aldrich 22092) supplemented with 50 µg/ml kanamycin (Sigma-Aldrich 60615) as needed.

## Mammalian cell culture

Experiments were performed with the murine macrophage cell line RAW264.7 (ATCC TIB-71TM). Macrophages were maintained at 37˚C with 5% $CO_2$ in Dulbecco's Modified Eagle Medium + GlutaMAX (DMEM, Gibco 61965–026) supplemented with 10% heat-inactivated fetal calf serum (FCS, Gibco 10270).

## Determination of MICs using E-test

Minimum inhibitory concentrations (MICs) were determined using standard test strips for ciprofloxacin (CIP, Liofilchem 920451), doxycycline (DOX, Liofilchem 921561), rifampicin (RIF, Liofilchem 920011) and streptomycin (STR, Liofilchem 921121) on *Brucella* agar plates (Sigma-Aldrich 18795) supplemented with 5% defibrinated sheep blood (Oxoid SB055). Bacteria were grown for 20 h at 37˚C and 100 rpm in TSB to mid-exponential phase. 80 µl of a $3.1 \times 10^8$ CFU/ml suspension were spread evenly on the agar plate and one to two E-test strips were placed on the agar. MICs were read after 3 days of incubation at 37˚C on the intersection of the even bacterial lawn with the inhibition ellipse at the test strip.

## RAW macrophage infection

Infections were performed in plastic (Corning 3904) or, for microscopy, glass-bottom (Greiner bio-one 655892) 96-well plates, for the reinfection assay 6-well plates (Falcon 353046) were used. The day prior to infection the indicated number of cells were seeded in respective plates. *B. abortus* strains were grown to an $OD_{600nm}$ of 0.8–1.1 (mid-exponential phase), diluted in DMEM/10% FCS to a multiplicity of infection (MOI) of 50 and added to the macrophages. Plates were centrifuged at 400 x g for 10 min at room temperature prior to incubation at 37˚C with 5% $CO_2$. The following steps are described individually for each experimental set-up (see below). Before fixation, all samples were washed 3 times with PBS to remove detached cells and free bacteria.

## *In cellulo* infection model to analyze inhibition of replication at different antibiotic concentrations (MIC$_{ic}$)

$1.5 \times 10^4$ RAW macrophages per well were seeded onto glass 96-well plates and infected as described above. At 1 h post infection (hpi) the infection medium was exchanged with DMEM/10% FCS containing 100 µg/ml gentamicin (GEN, Sigma-Aldrich G1397) and increasing concentrations of CIP (0, 0.078, 0.312, 1.25, 5, 20, 40 µg/ml, Sigma-Aldrich 17850), DOX (0, 0.1, 0.4, 1.6, 25, 100 µg/ml, Sigma D9891), RIF (0, 0.064, 0.32, 1.6, 8, 40, 100 µg/ml, Sigma R3501), STR (0, 0.64, 2, 3.2, 16, 80, 400 µg/ml, Sigma S9137) or indicated combinations. At 5 hpi the medium was exchanged with DMEM/10% FCS with 10 µg/ml GEN and respective

antibiotics. GEN was used in infection assays to avoid reinfection. Before fixation, all samples were washed 3 times with PBS to remove dead cells and bacteria. The cells were fixed 23 hpi and stained (see below). Imaging and CellProfiler analysis with readouts for integrated dsRed intensity and percentage of cells containing microcolonies were performed as described below.

### RAW macrophage infection to analyze transcriptional response in presence of antibiotics

$1x10^4$ cells per well were seeded onto glass 96-well plates and infected as described above. At 1 hpi the infection medium was exchanged with DMEM/10% FCS with 100 μg/ml GEN and 20 μg/ml CIP. At 5 hpi the medium was exchanged with DMEM/10% FCS with 10 μg/ml GEN and 20 μg/ml CIP. 23 hpi the medium was replaced with DMEM/10% FCS containing 100 ng/ml aTc as inducer. 27 hpi cells were washed 3x with PBS, fixed, and stained as described below. Imaging and CellProfiler analysis with readout for GFP-positive infection sites were performed as described below. Validation of the induction is presented in S1D–S1F Fig.

### Fixation and staining of infected cells for microscopy

Samples were fixed with 3.7% para-formaldehyde (PFA, Sigma, F1635) in 0.2 M HEPES (pH 7.2–7.4, Gibco 15630080) for 20 min at room temperature, washed three times with PBS and cell nuclei were stained with DAPI (1 μg/ml in PBS, Sigma-Aldrich D9542) for 15 min at room temperature. Stained samples were washed three times with PBS prior to imaging.

### Imaging

Microscopy was performed with Molecular Devices ImageXpress automated microscopes with a metaXpress plate acquisition wizard with gain 2, 12-bit dynamic range, for 25 sites per well in a 5x5 grid without spacing or overlap, with laser-based focusing and a 10x objective. Nuclei were imaged using DAPI, Bacteria were identified by dsRed and response to the inducer was measured in the GFP channel. The Site Autofocus was set to "All sites", and the initial well for finding the sample was set to "First well acquired". The Z-Offset for focus was selected manually and the exposure time was manually corrected to ensure a wide dynamic range with low exposure.

### Image analysis using CellProfiler

Images were analyzed using CellProfiler (version 2.2.0) [79]. A visual overview of the workflow is given in S2A Fig. Uneven illumination was corrected using an illumination function computed for every plate on all images based on the Background method. The resulting image was smoothed using a Gaussian method with 100-filter size.

The signal originating from *Brucella* DNA in the DAPI channel was reduced by subtraction of the dsRed image from the DAPI image. CellProfiler executed object segmentation and measurements with the following steps. (i) Nuclei stained with DAPI were detected as primary objects using the Adaptive strategy followed by the Maximum correlation thresholding method (MCT). (ii) Bacteria expressing dsRed or GFP were detected as primary objects using the Adaptive strategy followed by a Background method. Clumped objects were identified based on shape and segmented based on their intensity. For representative data of pipeline adjustments to distinguish clumped objects refer to S2B–S2D Fig (iii) To get an estimation of the cell body the nuclei were expanded by 3 pixels. (iv) To assign bacteria to specific cells the bacteria were masked with the expanded nuclei and bacteria partially or entirely masked were scored as 1. The number of infection sites and nuclei were extracted for each image. Area,

integrated intensity, and standard deviation of the intensity of the infection sites were measured. Potential artifacts were removed by setting a cut-off on the first quartile (Q1) of the dsRed intensity standard deviation. Data from all images from the same well were pooled and analyzed as a single condition. CellProfiler analysis resulted in one comma-separated values (csv) file per object category containing the selected measurements.

The typical area and typical diameter of wild-type *B. abortus* as well as mutant Δ*virB9*, a strain unable to replicate in infection due to the non-functional type 4 secretion system, were determined to set boundaries for microcolony identification (S3A–S3E Fig). Used read-outs were integrated dsRed intensity, percentage of cells containing microcolonies, and GFP-positive infection sites.

### Recovery of *B. abortus* from infected macrophages after antibiotic treatment

$2x10^4$ cells per well were seeded onto plastic 96-well plates and infected as described above. At 2 hpi the infection medium was exchanged with DMEM/10% FCS with 100 μg/ml GEN containing either CIP (20 μg/ml), DOX (100 μg/ml), RIF (100 μg/ml), STR (200 μg/ml) or indicated combinations. At 4 hpi the medium was again exchanged with DMEM/10% FCS with 10 μg/ml GEN together with the indicated antibiotics. At 4 and 24 hpi a subset of the infected macrophages was washed with PBS and lysed at room temperature for 10 min with pre-warmed 0.1% Triton X-100 (in water, Sigma-Aldrich T9284). Lysed cells were collected from the wells by pipetting up and down. Bacteria were collected by centrifugation of the lysates for 5 min at 16000 x g. The supernatant was discarded, and the pellets washed 2 times in PBS before resuspension in PBS. The recovered bacteria were plated onto TSA plates as described below. After 3 days of incubation at 37˚C the CFUs were counted, and the survival ratio was determined by division of the obtained CFU from the 24 h time point by the number of CFUs at 4 hpi.

### CFU plating of *B. abortus*

To obtain the number of colony forming units (CFUs) per ml, the *Brucella* resuspensions were serially diluted 1:5 in 96-well plates using PBS and 9 μl of each dilution were plated on square tryptic soy agar (TSA, Difco 236950) plates.

### Killing kinetics in the time-kill assay

*Brucella* grown to early- or mid-exponential phase ($OD_{600nm}$ 0.3 or 1.2, respectively) in TSB were diluted 1:10 in fresh TSB containing 10 μg/ml CIP and incubated at 37˚C and 100 rpm. At indicated time points samples were taken, centrifuged for 5 min at 16000 x g, pellet was washed 2x in PBS to remove antibiotics and CFUs were plated as described.

### Growth phase of bacteria vs. recovery after CIP treatment in broth

Bacteria were grown in TSB at 37˚C and 100 rpm. Samples were taken at indicated time points to enumerate CFU/ml by plating on TSA as described. At the same time samples were taken for subculturing in TSB containing 10 μg/ml CIP for 24 h at 37˚C and 100 rpm followed by CFU determination as described. The survival ratio was determined by dividing the obtained CFUs before and after treatment.

### Flow cytometry of *B. abortus*

*B. abortus* pAC042.8 was grown in TSB supplemented with 50 µg/ml kanamycin for 24 h at 37˚C and 100 rpm until $OD_{600nm}$ 1.4. Culture was then diluted 1:5000 or 1:10 in TSB or TSB containing 10 µg/ml CIP, respectively. Subcultures were incubated for 24 h at 37˚C and 100 rpm. GFP expression was induced by the addition of 100 ng/ml aTc for 4 h. Afterwards bacteria were pelleted (5 min, 16000 x g) and washed 2x in PBS. Samples were diluted 1:4 in 3.7% PFA and fixed for 30 min at room temperature. Following fixation, PFA was removed by centrifugation and pellets were resuspended in 0.22 µm filtered PBS. CountBright Absolute Counting Beads (Thermo Fisher C36950) were added to each sample for quantification of the bacteria. Fixed cells were subjected to flow cytometry using a BD LSR Fortessa. DsRed signal was acquired with a 561 nm laser (586/15 BP filter) and GFP with a 488 nm laser (505 LP Mirror and 512/25 BP Filter) and analyzed using FlowJo (FlowJo LLT, version 10.6.1). The total number of bacteria (dsRed+) and the proportion of metabolically-active (GFP+) bacteria were analyzed. Additionally, the CFUs were determined by dilution plating on TSA plates as described above.

### Persister formation during infection

Macrophages were infected with mid-exponential phase *B. abortus* (MOI 50). At 1 hpi the infection medium was exchanged with DMEM/10% FCS with 100 µg/ml GEN, to kill extracellular bacteria and prevent reinfection. At 5 hpi the infection medium was exchanged with DMEM/10% FCS with 10 µg/ml GEN. At indicated time points macrophages were lysed with 0.1% Triton X-100 for 10 min. Intracellular bacteria were recovered by pipetting up- and down and centrifugation for 5 min at 16000 x g. The pellet was washed two times in PBS before resuspension in TSB containing 10 µg/ml CIP. After 24 h at 37˚C and 100 rpm an aliquot of the culture was spun down and washed two times in PBS. Serial dilution plating was done to recover the CFUs/ml as described for samples before and after cultivation in supplemented TSB. The survival ratio was determined by division of CFUs recovered before by CFUs recovered after treatment.

### RAW macrophage reinfection assay

$3x10^5$ cells per well were seeded onto 6-well plates and infected as described above. At 1 hpi the infection medium was exchanged with DMEM/10% FCS with 100 µg/ml GEN. At 5 hpi the medium was exchanged with DMEM/10% FCS with 10 µg/ml GEN. At 24 hpi 20 µg/ml CIP was added to the infected cells. Samples were lysed with 0.1% Triton X-100 as described above at 3, 24, and 48 hpi to enumerate CFU/ml via CFU plating as described. Additionally, after 48 hpi a fraction of the recovered bacteria was subcultured in TSB at 37˚C and 100 rpm until $OD_{600nm}$ 0.9–1.1 and was used to start a new infection cycle as described. Data represent 5 independent infections and re-infections (i.e. 5 lineages). Infection-naïve *Brucella* were used in both rounds as control.

### Statistical analysis

Data aggregation was performed using KNIME (version 3.7.2) [80], visualization and analysis of the data were performed using TIBCO Spotfire Analyst (TIBCO, version 7.11.1) and/or GraphPad Prism 8.0.2. Modeling and statistical analysis were executed with GraphPad Prism. When applicable either paired t-test or ordinary one-way analysis of variance (ANOVA) with Tukey's multiple comparison test was performed as indicated in the figure legends.

Numerical data used to generate the figures and in statistical tests can be found in S1 Data.

## Results

### Efficacy and pharmacodynamics of mono- and dual-antimicrobial treatments on *B. abortus* in a macrophage infection model

Considering the proposed importance of the intracellular lifestyle of *Brucella* for its dissemination and maintenance within its hosts [25], we first investigated the influence of the intracellular state on the efficacy of different antibiotic treatments. Indeed, failure of the antibiotics used to treat brucellosis to reach and/or to kill intracellular *Brucellae* could be a critical factor contributing to the high relapse rate observed following antibiotic treatment of human brucellosis. To delineate the capacity of different antibiotics to reach phagocytosed *Brucella* and inhibit their growth, we infected the RAW264.7 macrophage cell line with *B. abortus* S2308 harboring a plasmid-encoded reporter system [78] (termed *B. abortus* here after). Infected cells were subjected to a range of antibiotic concentrations including 1 and 50 x the minimal inhibitory concentration (MIC), which we determined by classical E-Test on agar plates (Table 1).

We tested the efficacy of mono- as well as dual-treatments, as used in anti-brucellosis therapy, to inhibit intracellular bacterial replication. For this, macrophages were fixed, imaged, and analyzed using CellProfiler 2 [79] as described in detail in the method section. To define the intracellular minimum inhibitory concentration ($MIC_{ic}$) of the antibiotics tested in our experimental set up we used the integrated dsRed fluorescence of the bacterial reporter as a proxy for bacterial growth (Fig 1A–1G). Data was fitted with a Gompertz algorithm [85] using GraphPad Prism 8.0.2, and the associated $R^2$ value was used to assess the fitting accuracy (Fig 1A–1F). The $MIC_{ic}$s were defined at the crossing of the slope (red dotted line) and the bottom plateau (dotted black line) (Fig 1A–1F). The extrapolated $MIC_{ic}$s are compiled in Table 1, together with the antibiotic concentration typically reached in human serum upon treatment [81–84].

Mono-antibiotic treatments resulted in characteristic dose-dependent sigmoidal response curves (Fig 1A–1C), except for streptomycin (Fig 1D), which failed to inhibit intracellular growth of the bacteria. That finding is in line with the known inability of aminoglycosides to

**Table 1. MICs and $MIC_{ic}$s of *B. abortus* 2308 pAC042.8.**

| Antibiotic | MIC | $MIC_{ic}$ | | | $C_{max}$ |
|---|---|---|---|---|---|
| | µg/ml | µg/ml | SE | xMIC | µg/ml |
| CIP | 0.125–0.19 | 6.73 | 1.12 | 35–54 | 2.2 [81] |
| STR | 1–2 | >400 | - | >200 | 30–40 [82] |
| DOX | 0.125–0.25 | 6.09 | 1.27 | 24–49 | 2 [83] |
| RIF | 0.5–0.75 | 6.57 | 4.45 | 9–13 | 10 [84] |
| DOX + STR | | 1.58 3.16 | - | 6–13 2–3 | 2 [83] 30–40 [82] |
| DOX + RIF | | 1.64 1.64 | - | 6–13 2–3 | 2 [83] 10 [84] |

MIC = Minimal Inhibitory Concentration determined by E-test, n ≥ 3.

$MIC_{ic}$ = Minimal Inhibitory Concentration within macrophages extrapolated from the concentration-response curves (Fig 1), n = 3.

SE = standard error of $MIC_{ic}$.

xMIC = $MIC_{ic}$ value expressed as x times the MIC, values rounded.

$C_{max}$ = maximum reachable concentrations in human serum based on the indicated references.

CIP = ciprofloxacin, STR = streptomycin, DOX = doxycycline, RIF = rifampicin

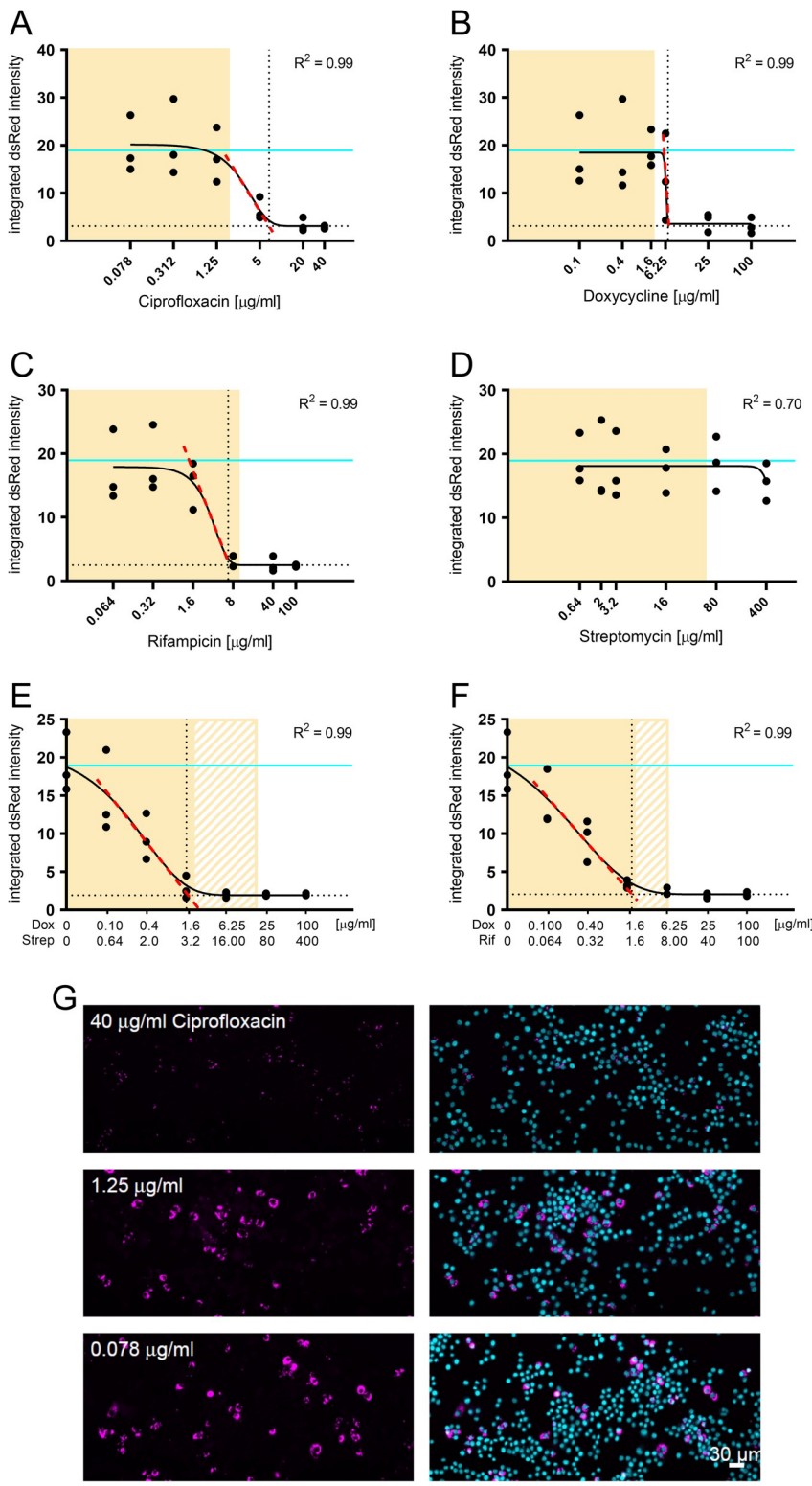

**Fig 1. Concentration-response curves of mono- and dual-therapy treatments for phagocytosed *B. abortus*.**
RAW264.7 macrophages were infected with *B. abortus* pAC042.8 for 1 h and treated with increasing concentrations of ciprofloxacin (A), doxycycline (B), rifampicin (C), streptomycin (D), doxycycline and streptomycin (E) or doxycycline and rifampicin (F) for 23 h. The concentrations were chosen to span 1x to 50x MIC. Bacterial load was determined by quantifying the integrated dsRed intensity per infected macrophage, extracted using CellProfiler and fitted with the

Gompertz algorithm to determine the $MIC_{ic}$ of all tested therapies (n = 3). Data points represent individual replicates. The fitting model is depicted as a black continuous line. The $MIC_{ic}$s were defined at the crossing of the slope (red dotted line) and the bottom plateau (dotted black line). Cyan lines represent the value measured in the control conditions without antibiotic treatment. Yellow shaded areas correspond to physiologically relevant antibiotic concentrations based on $C_{max}$ in human serum from Table 1. (E) and (F) solid yellow area correspond to physiologically relevant concentrations of doxycycline, whereas patterned areas correspond to physiologically relevant concentrations of streptomycin and rifampicin, respectively. (G) Representative images of infected RAW macrophages treated with increasing concentrations of ciprofloxacin. Bacteria constitutively expressed dsRed (magenta) and macrophage nuclei were stained with DAPI (cyan).

effectively reach intracellular bacteria [86]. At low antibiotic concentrations the integrated dsRed intensity in monotherapy samples (Fig 1A–1C) was comparable to the untreated control (cyan line), whereas dual-therapy samples already showed a decrease in intracellular replication at low antibiotic concentrations (Fig 1E and 1F). At high antibiotic concentrations (for both mono- and dual-therapies) the integrated dsRed intensity reached a plateau indicating cessation of intracellular bacterial growth on the population level.

For the monotherapies, only rifampicin treatment resulted in a physiologically relevant $MIC_{ic}$ (Table 1). Combination of doxycycline with either streptomycin or rifampicin, as recommended by the WHO, lead to significantly lower $MIC_{ic}$ when compared to monotherapies, resulting in physiologically relevant levels (Table 1). It is interesting to note the synergistic effect of streptomycin with doxycycline, although streptomycin alone did not show any measurable inhibitory effect (Fig 1D and 1E). All $MIC_{ic}$s obtained by this assay were above the obtained MICs on agar plates, supporting the protective nature of *Brucella*'s intracellular niche (Table 1).

## Antibiotic treatment fails to fully inhibit *B. abortus* replication within macrophages

A hallmark of *Brucella* intracellular replication is the formation of a so-called microcolony, resulting from the proliferation of the bacteria within their protective ER-derived niche [29, 36]. We thus analyzed the percentage of macrophages containing microcolonies from the previous infection data using a different CellProfiler pipeline (for details see methods). Treatment of infected macrophages with increasing concentrations of ciprofloxacin, doxycycline, and rifampicin lead to a gradual decrease in the percentage of microcolony containing macrophages (Fig 2A–2C). Streptomycin did not impair intracellular growth (Fig 2D), in concordance with our previous readout (Fig 1D). However, microcolonies were detected in all conditions, even at the highest antibiotic concentrations, albeit at low frequencies (Fig 2A–2F). To control that our pipeline did not misidentify microcolonies we used it to analyze *B. abortus* WT and *B. abortus* Δ*virB9* infected RAW264.7 macrophages as well as uninfected macrophages. *B. abortus* Δ*virB9* does not build a functional type 4 secretion system and is therefore microcolony-formation deficient [38]. Our pipeline identified no macrophages with microcolonies in uninfected and Δ*virB9* infected macrophages, whereas around 10% of *B. abortus* WT infected macrophages harbored microcolonies (Fig 2F). Additionally, we visually inspected the micrographs from the antibiotic treated samples and detected microcolonies even at high antibiotic concentrations (Fig 2E).

## A subpopulation of phagocytosed *B. abortus* remain physiologically active and cultivable following antibiotic treatment

Since high antibiotic concentrations and prolonged treatments (50x MIC for 24 h) did not seem to fully suppress intracellular replication (Fig 2), we assessed the physiological state of

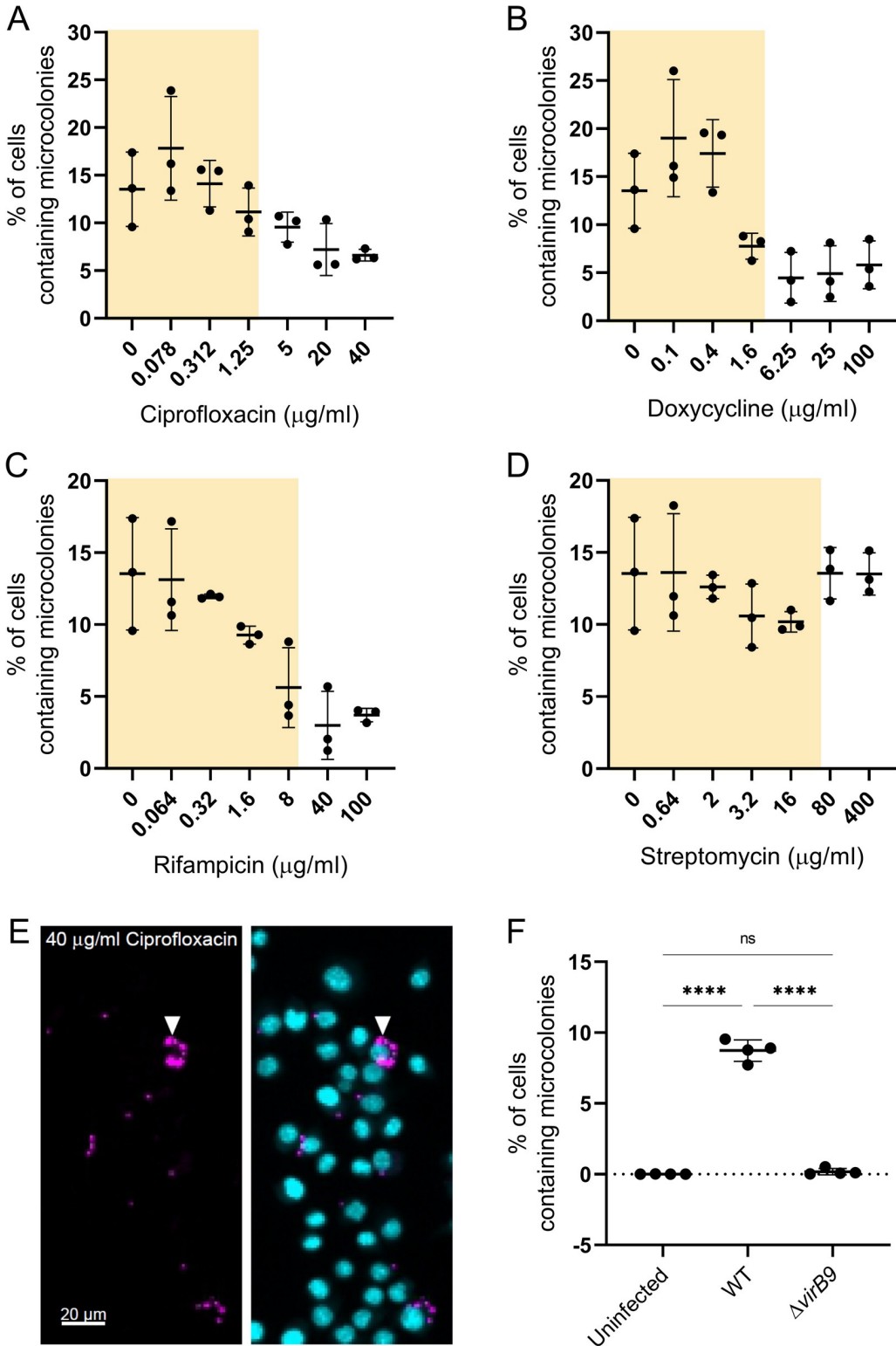

**Fig 2. Antibiotics are unable to fully inhibit *B. abortus* replication in macrophage infection.** RAW264.7 macrophages were infected with *B. abortus* pAC042.8 for 1 h and were then treated with increasing concentrations of ciprofloxacin (A), doxycycline (B), rifampicin (C), or streptomycin (D) for 23 h. Dots represent independent replicates, horizontal lines show the mean ±SD, n = 3. Yellow shaded areas correspond to physiologically relevant antibiotic concentrations based on $C_{max}$ in human serum from Table 1. (E) Representative image from (A). Bacteria constitutively expressed dsRed

(magenta) and macrophage nuclei were stained with DAPI (cyan). Arrowhead marks microcolony. (F) Macrophages either uninfected or infected with *B. abortus* WT (WT) or *B. abortus* ΔvirB9 (ΔvirB9) for 23 h were fixed and analyzed using our CellProfiler pipeline for percentage of cells containing microcolonies. Dots represent replicates, horizontal lines show the mean ±SD, n = 2 with 2x2 technical replicates (****$p \leq 0.0001$ with ordinary one-way ANOVA and Tukey's multiple comparison test).

antibiotic-treated intracellular *Brucella* by testing their ability to express a plasmid-encoded, tetracycline-inducible *gfp* reporter construct. To that end, we added aTc to the infected, antibiotic-treated macrophages, using ciprofloxacin treatment as a model (see methods for detail). A similar approach has already been extensively used to study intracellular *Salmonella* in a macrophage infection model [68, 87]. Strikingly, a remarkable subpopulation of the phagocytosed *Brucella* (about 2%) was able to produce GFP despite prolonged antibiotic treatment and were therefore considered transcriptionally and translationally active (Fig 3A and 3B).

Further, we tested the viability of phagocytosed *Brucella* by lysing and plating infected, antibiotic-treated macrophages using the same set-up as the one used in our previous assay. Following antibiotic treatment and macrophage lysis, a noticeable proportion of bacteria was able to resume growth on plate (Fig 3C). In concordance with our microscopy data, none of the tested antibiotics, not even the dual-therapies recommended by the WHO, lead to full sterility of the treated macrophages (Fig 3C). These results consolidate our previous findings that even after prolonged antibiotic treatment with high antibiotic concentrations some phagocytosed *Brucella* still formed microcolonies (Fig 2). Of note, the proportion of metabolically-active bacteria (1.97 ± 0.98%) was markedly higher than the proportion of viable bacteria recovered after treatment with the same antibiotic(s) in our plating experiment (0.12 ± 0.14%), suggesting that not all metabolically-active *Brucella* could resume growth under the tested conditions. These

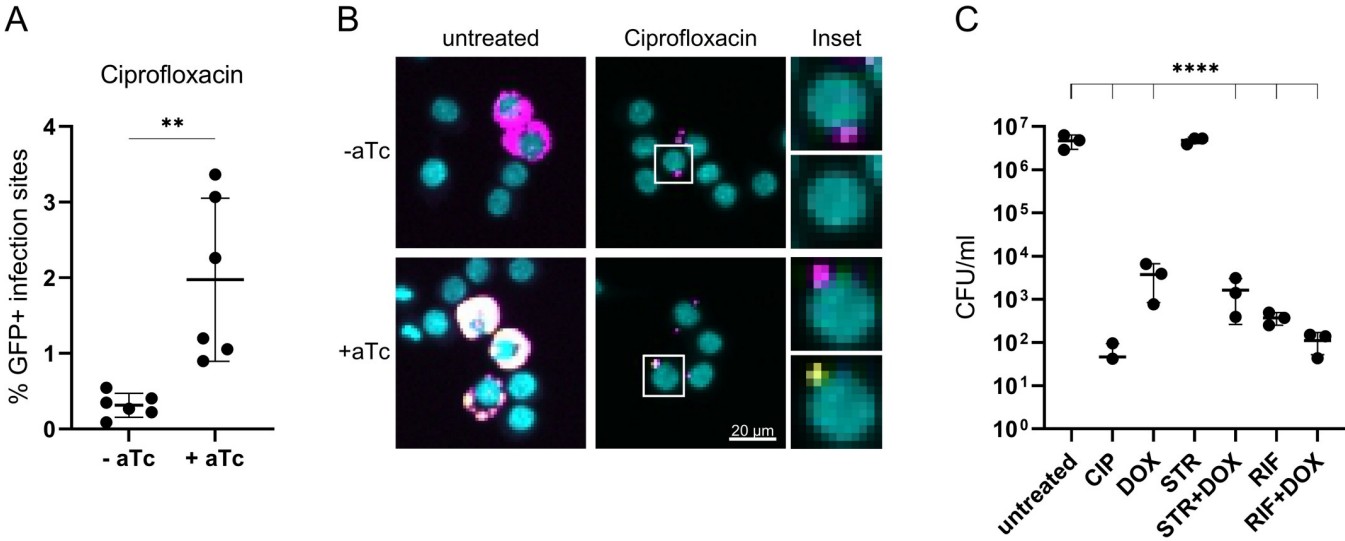

**Fig 3. A subpopulation of phagocytosed *B. abortus* is viable and virulent after antibiotic treatment.** (A, B) RAW264.7 were infected with *B. abortus* pAC42.08 for 1 h and then treated with 20 µg/ml ciprofloxacin. At 23 hpi aTc was added. Infected macrophages were fixed 27 hpi prior to imaging and CellProfiler analysis. (A) Infection sites denotes any dsRed+ object identified by our CellProfiler pipeline. Dots represent independent replicates, horizontal lines show the mean ±SD, n = 6 (**$p \leq 0.083$ with paired t-test). (B) Bacteria constitutively expressed dsRed (magenta) and macrophage nuclei were stained with DAPI (cyan). GFP (yellow) was expressed from an aTc inducible plasmid. Insets show split channels for dsRed (top) and GFP (bottom). (C) RAW264.7 were infected with *B. abortus* pAC42.08 for 2 h and then treated with ciprofloxacin (CIP, 20 µg/ml), doxycycline (DOX, 100 µg/ml), rifampicin (RIF, 100 µg/ml), streptomycin (STR, 200 µg/ml) or indicated combinations. At 24 hpi macrophages were lysed and intracellular bacteria were plated. TSA plates were incubated at 37˚C for 3–4 days when CFU/ml were enumerated. Data shows individual replicates with n ≥ 3, horizontal lines show the mean ±SD (****$p \leq 0.0001$ with ordinary one-way ANOVA and Tukey's multiple comparison test). Y-axis in log-scale.

results show that in our macrophage infection model a subpopulation of phagocytosed *B. abortus* remain metabolically-active and cultivable following antibiotic treatment.

### *Brucella* forms "persisters" both in broth and within infected cells

Antibiotic persisters have been defined as a sub-population of bacteria that survives repetitive exposures to bactericidal antibiotic treatments above the MIC without a resistance mechanism and largely independent of the type of antibiotic used [73]. A bi-phasic curve, with a slower killing rate of the persistent population in comparison to the susceptible population, in a time kill assay is a signature of bacterial "persisters" [73]. We were intrigued whether such a phenomenon could be linked to the intracellular survival under antibiotic treatment we observed in our experiments (Fig 3).

Following reports from Amraei *et al.* (2020) that *B. abortus* (B19) and *B. melitensis* (16M) display biphasic killing curves in broth in presence of ampicillin and gentamicin [61, 62], we tested whether this could also be observed for ciprofloxacin treatment of *B. abortus*. To this end, *B. abortus* liquid cultures were treated with ciprofloxacin for 1 to 48 h followed by CFU determination after removal of the antibiotic (Fig 4A). To test a possible influence of the density of the treated culture [73] in our time-kill assay we compared early- ($OD_{600nm}$ 0.3) and mid-exponential ($OD_{600nm}$ 1.2) cultures. The survival ratio was determined by dividing CFUs obtained from the culture before and after treatment with ciprofloxacin. In the first 6 h the number of viable bacteria decreased fast, which was followed by a slower killing rate (Fig 4A), resulting in a characteristic bi-phasic curve. There was no observable difference in survival between the two tested bacterial growth conditions in our killing assay, ruling out a strong effect of the starting density of the culture (Fig 4A). To further investigate the influence of *Brucella* growth on antibiotic susceptibility and on the presence of these "persister" cells we determined survival ratios at different stages of growth using 50 µg/ml ciprofloxacin as treatment (saturated killing, Fig 4B). Survival appeared to be strongly dependent on the growth phase of the bacteria as the survival ratio kept increasing once the culture reached stationary phase (Fig 4B), reaching about 10% of the entire population after 3 days of culture. We also noticed an initial decrease in the survival ratio (between 1 h and 24 h), which we attributed to the dilution of the persisters introduced in the cultures at the inoculation, as we routinely inoculate from stocks of stationary phase bacteria (see methods).

Further, we assessed the physiological state of ciprofloxacin-treated *B. abortus* grown in broth, using the beforementioned reporter system. We wanted to compare the physiological state of ciprofloxacin-treated *B. abortus* grown in broth with the physiological state of phagocytosed, ciprofloxacin-treated *B. abortus* (Fig 3). To this end, we treated *B. abortus* pAC042.8 grown in broth for 24 h with ciprofloxacin. After 24 h treatment we induced GFP expression by addition of aTc for 4 h. Fixed samples were used to identify the total number of bacteria (dsRed+) and the proportion of metabolically-active bacteria (GFP+) using flow cytometry (S4A and S4B Fig). Additionally, CFUs were determined by dilution plating on TSA plates to assess the proportion of viable bacteria after treatment (Fig 4C). Whereas 0.94% (± 0.6%) of the ciprofloxacin-treated bacterial population was responsive to the inducer aTc (GFP+), only 0.01% (± 0.005%) resumed growth after removal of the antibiotic pressure (Fig 4C). These results corroborated our findings with infected macrophages, both on the presence of a metabolically-active sub-population of bacteria despite high antibiotic concentration, and that only a fraction of these bacteria could resume growth under our tested conditions.

Ciprofloxacin treatment in broth seemed to be more efficient compared to treatment of phagocytosed bacteria highlighting once again the protective nature of the intracellular niche (s). As intracellular conditions vary in many physico-chemical parameters [88] and are much

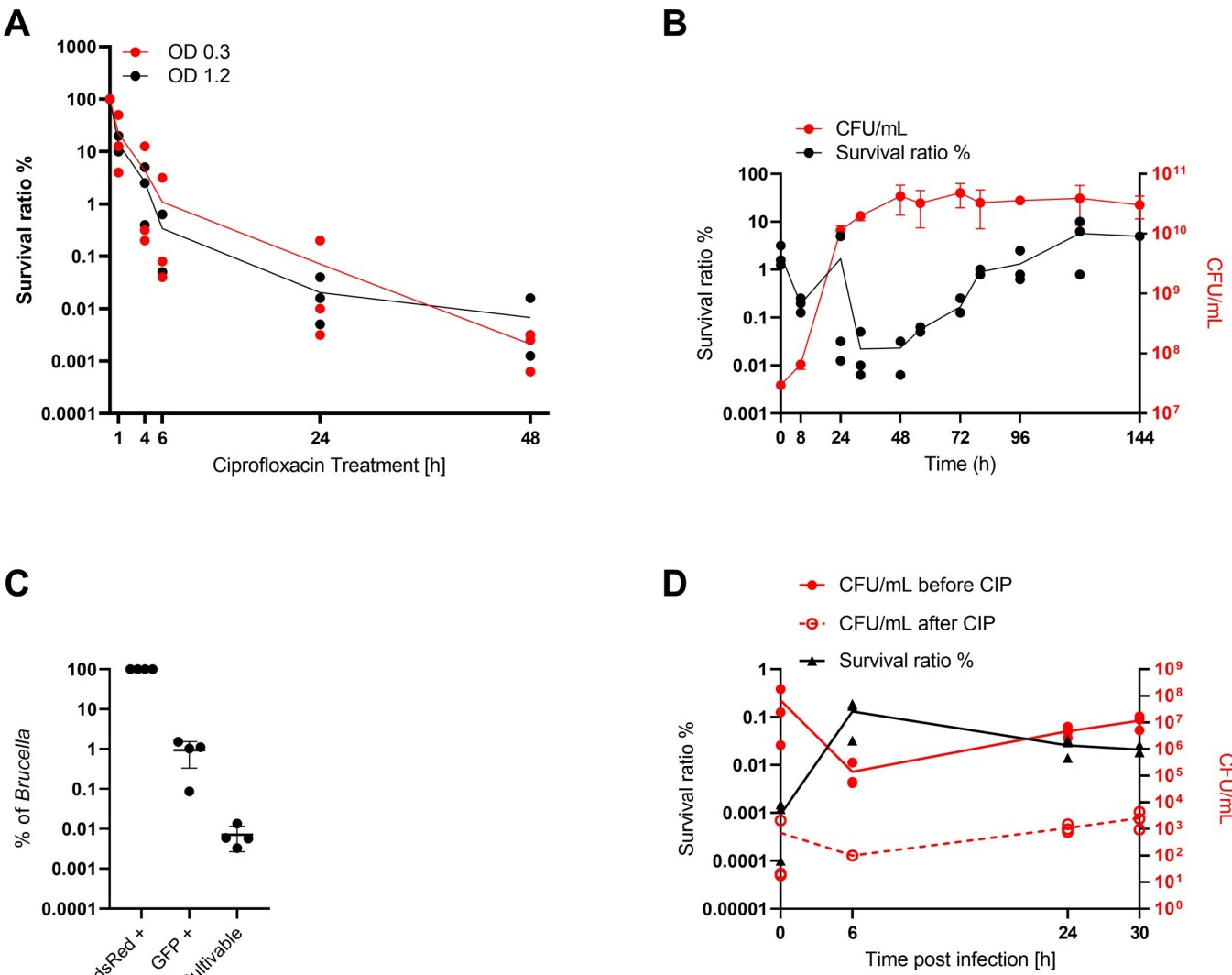

**Fig 4. *B. abortus* forms ciprofloxacin-tolerant subpopulation in broth and in infection.** (A) Killing kinetics of *B. abortus* cultures OD$_{600nm}$ 0.3 (red) or 1.2 (black) treated with 10 µg/ml ciprofloxacin. Lines = average survival ratio. Dots represent independent experiments, n = 3. Y-axis in log-scale. (B) Bacteria were grown in TSB, samples were taken at indicated time points to enumerate CFU/ml by plating on TSA and for subculturing in TSB containing ciprofloxacin for 24 h followed by CFU determination. The survival ratio was determined by dividing the obtained CFUs before and after treatment. Displayed are the individual data points and the average for the survival ratio (black) as well as the average and the associated standard deviation for CFU/ml (red), n = 3. Both Y-axis in log-scale. (C) An exponential *B. abortus* culture treated for 24 h with ciprofloxacin was incubated with aTc for 4 h. Samples were fixed and analyzed using flow cytometry to identify the total number of bacteria (dsRed+) and the proportion of metabolically-active bacteria (GFP+). The number of cultivable bacteria was assessed on plate following dilution plating from the same culture. Data represent the mean ± SD, dots represent independent experiments, n = 4. Y-axis in log-scale. (D) RAW264.7 macrophages were infected with mid-exponential phase *B. abortus* (MOI 50). At indicated time-points macrophages were lysed, intracellular bacteria recovered and sub-cultured in ciprofloxacin-supplemented TSB. Serial dilution plating to recover the CFUs/ml were done before (red line) and after (red dotted line) cultivation in CIP supplemented TSB. The survival ratio (black line) was determined by division of CFUs recovered before by CFUs recovered after treatment. Dots represent independent experiments, n = 3. Both Y-axis in log-scale.

more complex compared to growth in rich medium we wanted to investigate if *de novo* "persister" formation could also be observed following macrophage infection. Indeed, phagocytosis of *Salmonella* leads to the enrichment and to the formation of such a transiently non-growing antibiotic tolerant subpopulation [67–69]. To assess the proportion of persisters formed following phagocytosis of *Brucella*, we infected RAW264.7 macrophages with *B. abortus* wild-type in a gentamicin-protection assay set up (see methods). Macrophages were lysed at

indicated time-points and intracellular bacteria were recovered. A fraction of the bacteria was plated on TSA for CFU determination, and the remaining bacteria were sub-cultured in medium containing ciprofloxacin for 24 h before CFU determination. To determine the persister frequency, the ratio of recovered CFU before (Fig 4D red line) and after (Fig 4D red dotted line) antibiotic treatment was calculated (Fig 4D black line). Between 0 h and 6 h a decline in total CFU (untreated) was observed, which is attributable to killing of the non-phagocytosed bacteria by gentamicin (extra-cellular killing) and killing of a fraction of phagocytosed bacteria by the bactericidal innate immune response of the macrophage (Fig 4D). Between 6 and 24 h the phagocytosed bacteria started to replicate leading to an increase of CFUs. During the first 6 h an approximately two log increase in the persister frequency was observed (0.0009% to 0.13%, Fig 4D). From 6 to 30 h the absolute number of persisters (Fig 4D) increased although the persister ratio decreased since phagocytosed bacteria started to replicate. This finding indicates that persisters could be formed *de novo* during macrophage infection. The observed phenomenon was not attributable to the experimental set-up, as contact to macrophages, Triton X-100 treatment or contact to the infection medium did not increase the persister ratio (S4C Fig).

### The subpopulation of intracellular bacteria surviving antibiotic treatment remains virulent

Finally, we assessed whether the intracellular subpopulation surviving antibiotic treatment retained the ability to seed a new infection. To this end we regrew bacteria from ciprofloxacin-treated macrophages (Fig 5, filled symbols) and used these to infect fresh macrophages (Fig 5, empty symbols). We monitored the infection by lysing and plating a fraction of the macrophages at 3 hpi to determine uptake by the macrophages and at 24 hpi to determine intracellular replication (Fig 5, filled symbols). The remaining infected macrophages were treated for an additional 24h with ciprofloxacin, before lysis and CFU determination (Fig 5, filled symbols). Part of the bacteria recovered at 48 hpi were subcultured in liquid broth and used to start a new infection (Fig 5, empty symbols). For both infection cycles the original *B. abortus* strain was used as control (Fig 5, red symbols). We did not observe any difference in the amount of recovered CFUs after antibiotic treatment from the passaged strains compared to a naïve infection (Fig 5), indicating that the subpopulation surviving antibiotic treatment remained susceptible to ciprofloxacin. Importantly, these bacteria stayed virulent, as shown by their ability to seed a new infection.

In summary, we confirmed that a sub-population of *Brucella* grown in broth under optimal conditions does present the characteristics of so-called "persisters", and that *de novo* persister formation is also taking place during macrophage infection. Furthermore, the surviving subpopulation recovered from treated macrophages remains fully virulent and is able to seed a new infection.

## Discussion

Persistent or chronic infections constitute an insidious medical challenge and lead to a high socio-economic burden on the affected individuals and populations. Although human brucellosis is rarely lethal, its occurrence is associated with a high morbidity, and the underlying causes of the observed clinical failure remain largely elusive [21, 50, 51]. Prevalence is mostly restricted to low-income regions and treatment options have not evolved for decades. First line antibiotics, like doxycycline, streptomycin, and rifampicin remain potent against *Brucellae*, when tested *in vitro*, and genetic resistance does not seem a major threat, in contrast to what is observed for many other human pathogens. However, the relatively high relapse rate

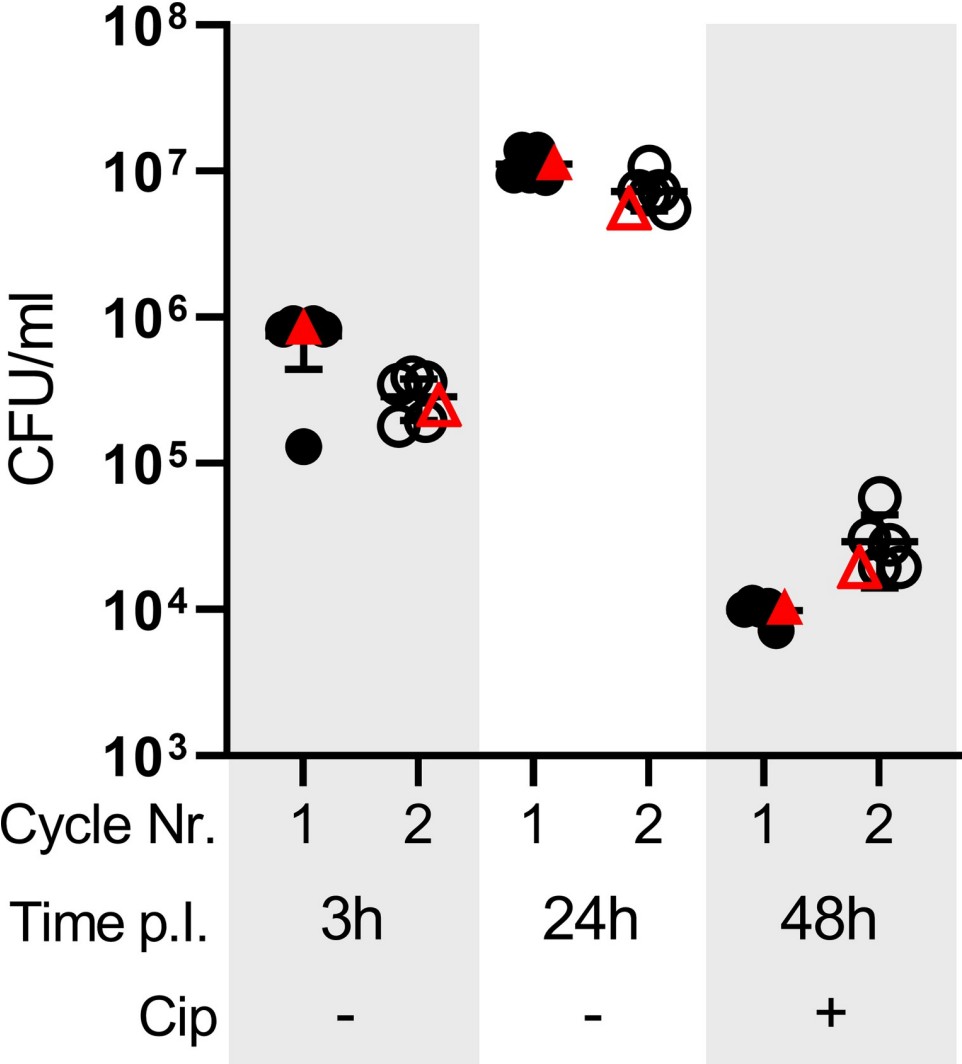

**Fig 5. Phagocytosed *B. abortus* surviving antibiotic treatment remain virulent.** RAW264.7 were infected with infection-naïve *B. abortus* or bacteria regrown from antibiotic-treated macrophages. Ciprofloxacin was added at 24 hpi. Samples were lysed at 3, 24 and 48 hpi to enumerate CFU/ml. Additionally, after 48 hpi a fraction of the recovered bacteria were subcultured in TSB and after regrowth were used to start a second infection. Black symbols represent independent bacterial lines, red symbols depict infection-naive *Brucella* as control, filled symbols show first cycle, empty symbols show second cycle, horizontal lines show the mean ±SD. Y-axis in log-scale.

contributes to the high disease burden of human brucellosis. Whereas re-infection or non-compliance to the prescribed treatment may account for a fraction of the observed cases [58], it is manifest that other reasons are involved, which are linked to the infectious properties of the agent.

## A protective intracellular state

Our macrophage infection model unambiguously demonstrated the protective nature of *Brucella's* ability to survive intracellularly. Indeed, all the antibiotic treatments we tested (single and dual treatments) failed to completely inhibit intracellular replication, and *a fortiori* to reach sterility (Fig 2). Considering that intracellular replication only starts between 5 and 10

hpi [36], while the treatment started already at 1 hpi, the finding may indicate that a fraction of the intracellular bacteria were able to reach their replicative niche and replicate despite the presence of the different antibiotics. The protective nature of *Brucella's* intracellular state could also result–at least in part–from a possible phenotypic heterogeneity in the macrophage population in respect to antibiotic uptake and/or efflux. Interestingly, Czyz *et al.* found that none of 480 tested bioactive compounds against intracellular *B. abortus* [89] lead to sterility in a macrophage infection model although some of the tested compounds inhibited replication *in cellulo*. Using an extracellular vs. intracellular MIC comparison (Table 1), we determined that the protective effect of the intracellular state ranged from about 10-fold for rifampicin to roughly 50-fold for ciprofloxacin and doxycycline. Most concerning, all obtained $MIC_{ic}$ values —except that of rifampicin—were above physiologically relevant antibiotic concentrations for monotherapies, correlating with the knowledge that such treatments are linked with high relapse rates [59] and are therefore not encouraged. Although monotherapy with rifampicin theoretically may be sufficient, its single use should be avoided as clinical resistance has already been reported [54–57, 90]. While our $MIC_{ic}$ results globally support the recommended dual therapies with inhibition being observed at physiologically relevant concentrations (Figs 1–2), they nevertheless did not sterilize. To overcome a potential protective effect of the *Brucella* intracellular niche(s), antibiotics could be loaded onto nanocarriers or into nanoparticles. In this line, Bodaghabadi and colleagues aimed for enhancing uptake of rifampicin into *Brucella*-infected macrophages by loading nanocarriers with rifampicin. However, treatment with rifampicin loaded nanocarrier did not lead to significantly lower bacterial loads than treatment with free rifampicin [91]. In another study Imbuluzqueta and colleagues encapsulated hydrophobically modified gentamicin in poly(lactic-co-glycolic acid) (PLGA) nanoparticles which resulted in improved efficacy *in cellulo* and also *in vivo* [92].

The protectiveness of intracellular niches against antibiotics is a common concern for the treatment of intracellular pathogens. Indeed, protective effects have been observed for *Legionella pneumophila*, *Coxiella burnetii*, *Mycobacterium tuberculosis*, and uropathogenic *Escherichia coli* among others. For *L. pneumophila* which alike *Brucella* is associated to an ER-derived intracellular niche [93] the observed intracellular MIC is roughly 2 and 17 times the extracellular MIC for ciprofloxacin and rifampicin, respectively [94], whereas no difference is observed for doxycycline or chloramphenicol [94]. For *C. burnetii*, which survives in a highly acidified phagolysosome-like compartment [95, 96], a similar effect was described for doxycycline and ciprofloxacin with a protective range spanning 2-5x MIC [97, 98]. Similar results were obtained with ciprofloxacin and other quinolones against intracellular *Listeria monocytogenes* (localized in the cytosol, 3x MIC), *Staphylococcus aureus* (localized in the phagolysosome, 8x MIC), and uropathogenic *E. coli* (localized in acidified compartment, >3x MIC) compared to extracellular bacteria [99, 100]. In the case of *M. tuberculosis*, which can reside in different cellular compartments such as phagosomes and autophagosomes (reviewed in [101]), the protective effect of the intracellular niche is about 100-fold for rifampicin when compared to extracellular growth [102, 103].

Whereas these studies individually confirm the general protective nature of intracellular lifestyles against antibiotic treatments, such as we describe in the present study, it is very interesting to note that there are some striking discrepancies when looking at the results of individual antibiotics in relation to the pathogens and their diverse niches. For instance, the protectiveness of *Brucella*'s ER-associated intracellular niche against chloramphenicol has already long been recognized [104], although this antibiotic readily kills intracellular *H. influenzae* and *L. pneumophila* [94, 105]. Inversely, if *Brucella*'s and *Coxiella's* [98] intracellular niches seem to confer a substantial protection against doxycycline, the one of *L. pneumophila's* fails to do so [94]. That the low pH in *Coxiella's* phagolysosomal niche is most likely one of the main factors conferring partial protection against different antibiotics is a well-recognized fact

[106–109]. However, *B. abortus*, *L. pneumophila* and *M. tuberculosis* replicate in non-acidified compartments (reviewed in [110]). The combination of pathogen-specific factors and the nature of the intracellular niche(s) hence constitute additional layers of complexity when seeking for broad-range antimicrobials active *in cellulo* and eventually *in vivo*.

## A protective *persister* state

Considering that even the harshest treatments failed to eradicate *Brucella* in our infection model, we investigated whether this resilience could be linked to the "persister" phenomenon, a transient phenotypic state resulting in multi-drug tolerant bacterial subpopulation(s). Research on persisters has recently regained a broad attention as accumulating evidence has shed light on the clinical relevance of this intrinsic bacterial property to survive prolonged exposure to bactericidal treatments. For instance, persisters have been recognized as a challenge to control or neutralize chronic-infection causing bacteria like *Salmonella*, *M. tuberculosis*, *Pseudomonas aeruginosa*, and *S. aureus*, and have been identified as one of the underlying reasons for chronic and recurrent infections, lengthy therapies, and relapses [111–115]. These pathogens can convert a part of their population to a slow- or non-replicative state to persevere the stress of the host cell environment—with the added benefit for the pathogen that these physiological changes also reduce susceptibility to antibiotics (reviewed in [116, 117]).

Our present report confirms the occurrence of this phenomenon for *Brucella* in broth, as recently published [61, 62], and extends it to *in cellulo*, as shown in our macrophage infection model. In contrast to *Salmonella* and *M. tuberculosis*, for which persister formation has been reported to be induced by macrophage infection [64, 65, 68, 118], our experiments did not allow to conclude whether the internalization and subsequent replication in these cells triggered or rather enriched for persisters. However, their presence was clearly demonstrated, with a possible *de novo* formation during intracellular replication (Fig 4D). Having demonstrated that the regrowth of these intracellular persisters regenerated a fully infective population (Fig 5), it is very tempting to speculate that intracellular *Brucella* persisters contribute to the establishment of chronic brucellosis and to the observed resilience of the pathogen to antibiotic treatment by constituting a potent reservoir for re-infection. In that perspective, identifying the mechanisms underlying the formation of such sub-population(s), their associated physiological states, the exact nature of their intracellular niche(s), and assessing their possible involvement in the development of a systemic infection will constitute an important future research direction, as will be the investigation of persisters in clinical samples from human patients. Our findings further support the need to include anti-persister molecules in future treatment development, a field that is in full expansion. Promising candidates have already been described. For instance, ADEP4 and lassomycin, active against *S. aureus* [119] and *M. tuberculosis* [120] interact with the protease ClpP leading to a direct killing of the persister cells. Combining ADEP4 and rifampicin lead to the complete eradication of *S. aureus* stationary population and biofilm [121], whereas Lassomycin cleared *M. tuberculosis* persisters in stationary phase culture [120]. Other drugs, such as Tridecaptin M, which sensitizes *Acinetobacter baumanii* persisters for killing by rifampicin [122] or 4-hexylresorcinol (4-HR), a phenolic lipid, which potentiated the MIC of various classes of antibiotics up to 50-fold for different bacterial species such as *Klebsiella pneumoniae*, *S. aureus* and *P. aeruginosa* [123] demonstrate that persisters can specifically be targeted. Another mechanism to improve the treatment of relapsing infections was demonstrated recently for *Salmonella* by Li, Claudi, Fanous et al. [124]. They showed that eradication of *Salmonella* from infected host tissue in a mouse model of infection was only possible after addition of adjunctive immune therapy, highlighting the relevance of the immune system in eradication of persistent infections.

## Conclusion

In this study, we show the advantages of a simple *in cellulo* model over classical *in vitro* approaches in broth for testing antimicrobials against *Brucella*. However, even if reflecting the conditions met in human better, our model is far from capturing the diversity of the host microenvironments encountered by the pathogen during the course of an infection [125]. Future model refinements and diversification will be necessary to tackle the problem caused by relapses in human brucellosis. Together, our results highlight the need to extend the spectrum of models used to test new antimicrobial therapies for brucellosis, including a special consideration for compounds targeting the persister population.

## Supporting information

**S1 Fig. Response of *B. abortus* pAC042.8 to inducer in broth and in RAW macrophage infection.** (A) Schematic representation of plasmid pAC042.8. Bacteria carrying pAC042.8 constitutively express dsRed and express GFP under a tetracycline-inducible system. (B) Flow-cytometric identification of the bacterial population grown till the exponential phase based on its dsRed fluorescence and forward-scatter (FSC) property. (C) Flow-cytometric detection of the GFP fluorescence in the bacterial population grown till exponential phase (magenta), after the addition of the inducer (green) and 24 h after removal of the inducer (yellow). For (B) and (C), displayed is a representative dot blot of one experiment of at least 3 independent replicates. (D, E, F) RAW macrophages were infected with *B. abortus* pAC042.8 or *B. abortus* Δ*virB9* pAC042.8 for 27 h. Cells were treated with aTc (100 ng/ml) at 23 hpi for 4 h before fixation and imaging. (D) Box plot showing the size-independent response of the bacterial population to the inducer (**** $p \leq 0.0001$; ns = non-significant, one-way ANOVA and Tukey's multiple comparison test). (E) Fold induction of the bacterial population response to the inducer. Fold induction was calculated by dividing the median values of the induced population by the non-induced one between matching conditions. Each dot represents one independent replicate, the horizontal line represents the mean, n = 2. (F) Percentage of GFP+ infection sites. Horizontal bars represent the mean, n = 2.
(TIF)

**S2 Fig. Visual overview of CellProfiler workflow and method to distinguish clumped objects.** (A) Image analysis workflow for CellProfiler (https://cellprofiler.org) for the segmentation of nuclei and infection sites. The minimum typical diameter was set to different values in the pipelines to either identify non-replicative infection sites or microcolonies. Nuclei were expanded by 3 pixels and only infection sites residing inside the expanded nuclei were kept for the analysis. Created with BioRender.com. (B) Representative images from RAW264.7 macrophages infected for 27 h with *B. abortus* pAC042.8 constitutively expressing dsRed (magenta). Macrophage nuclei were stained with DAPI (cyan). Image analysis was performed with Cell-Profiler to segment nuclei and bacteria and to extract measurements. The left panel shows microcolonies segmentation with a typical diameter set between 1 and 30 pixels. The middle and right panels show microcolonies segmentation with a typical diameter between 5 and 30 pixels, without (middle) clumped-objects correction. The right panel shows microcolonies segmentation after identification of clumped objects based on the shape followed by the division of those based on the intensity. Arrows indicate examples of separation of merged microcolonies. (C) Percent error between manual count and CellProfiler count using a typical diameter between 1 and 30 or between 5 and 30. Data represent the mean ±SD, n = 10 (5 x 2 sites per experiment). (D) Violin plot showing the distribution of the percent error between manual count and CellProfiler count of microcolonies using several methods to distinguish and

separate clumped objects. The pink line represents the median, dotted lines represent quartiles. n = 10 (5 x 2 sites per experiment).
(TIF)

**S3 Fig. CellProfiler pipeline set-up to define infection site sizes.** (A) Representative violin plot depicting the distribution of the standard deviation of the dsRed intensity for each infection site of *B. abortus* wild-type (WT), Δ*virB9* and wild-type killed with gentamicin for 24 h (GEN-killed) all carrying pAC042.8. Horizontal blue lines display first and third quartiles, horizontal black dotted line represents median. Red box shows remaining event after filtration with CellProfiler pipeline. n = 2 (B) Area of infection sites before intracellular replication. The 90th percentile (P90) of the measured area for each condition after 5 h infection. Horizontal lines represent the average, n = 4 (2 x 2 technical replicates, ns = non-significant with paired t-test). (C) Average area and typical diameter of infection sites of *B. abortus* wild-type (WT) and Δ*virB9* at 5 hpi. Table shows the equivalence between the measured area and the diameter of a circle using the formula $dd = 2(\sqrt{A}/\sqrt{\pi})$. Displayed are the averages from 2 technical replicates from 2 independent experiments. (D) Representative images from RAW264.7 infected for 6 h or 27 h with *B. abortus* WT or Δ*virB9*. Bacteria constitutively expressed dsRed (magenta) and macrophage nuclei were stained with DAPI (cyan). (E) Entry rate from macrophages infected with *B. abortus* WT or *B. abortus* Δ*virB9* carrying plasmid pAC042.8 at 5 hpi. The entry was determined using the number of nuclei associated with at least one non-replicative infection site in the 3 pixels-expanded nuclei area. Horizontal lines show the average, n = 2 (2 x 2 technical replicates) ($^{**}p \leq 0.01$, $^{***}p \leq 0.001$, ns = non-significant, one-way ANOVA and Tukey's multiple comparison test).
(TIF)

**S4 Fig. GFP-induction in broth, control treatments for macrophage lysis, and pre-dilution of inoculum.** (A) Comparisons between CFU counts and flow cytometric counts using counting beads. Data represent the mean ± SD, n = 4 (ns = non-significant, paired t-test). (B) Flow-cytometric identification of the GFP+ population. Representative dot blots of one experiment of 3 independent replicates. (C) *B. abortus* was grown to mid exponential phase, macrophages were infected with *B. abortus* for 10 min, and *B. abortus* was incubated for 10 min in Triton X-100. After each step samples were washed, and one part of the sample was used to enumerate CFU/ml before ciprofloxacin treatment. Another part of each sample was subcultured in TSB plus ciprofloxacin and after 24 h CFU/ml were enumerated. The survival ratio was calculated by division of CFUs recovered before by CFUs recovered after ciprofloxacin treatment. (ns = non-significant, one-way ANOVA and Tukey's multiple comparison test).
(TIF)

**S1 Data. S1 Data comprises all numerical values used to generate the figures presented in this manuscript, with the dataset corresponding to individual figure panels placed in separated tabs.**
(XLSX)

## Acknowledgments

We thank Julie Sollier for critically reading of the manuscript. We are grateful to Sonia Borrell, Miriam Reinhard, Anna Dötsch, Ainhoa Arbues Arribas and colleagues from the Swiss TPH, Basel, Switzerland, for their support and assistance in the BSL-3. We also thank Therese Tschon for excellent assistance on *Brucella* specific work in and outside the BSL-3. Special thanks to Kai Schleicher and colleagues from the Imaging Core Facility of the Biozentrum,

Basel, Switzerland, for extensive assistance with microscopy and Cell Profiler analysis, to Janine Bögli from the FACS Core Facility of the Biozentrum, Basel, Switzerland, for help with FACS sample processing and analysis. Finally, we want to thank Dr. Sophie Helaine for constructive advice during the development of this project.

## Author Contributions

**Conceptualization:** Selma Mode, Maxime Québatte, Christoph Dehio.

**Data curation:** Selma Mode, Maren Ketterer, Maxime Québatte.

**Formal analysis:** Selma Mode, Maren Ketterer, Maxime Québatte.

**Funding acquisition:** Christoph Dehio.

**Investigation:** Selma Mode, Maren Ketterer, Maxime Québatte.

**Methodology:** Selma Mode, Maren Ketterer, Maxime Québatte.

**Project administration:** Selma Mode, Maren Ketterer, Maxime Québatte, Christoph Dehio.

**Supervision:** Maxime Québatte, Christoph Dehio.

**Validation:** Selma Mode, Maren Ketterer, Maxime Québatte.

**Visualization:** Selma Mode, Maren Ketterer.

**Writing – original draft:** Maren Ketterer, Maxime Québatte.

**Writing – review & editing:** Maren Ketterer, Maxime Québatte, Christoph Dehio.

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
