## [Decision Letter · Decision Letter 0]

24 May 2022

Dear Dr. Dehio,

Thank you very much for submitting your manuscript "Antibiotic persistence of Brucella abortus in its protective intracellular niche" for consideration at PLOS Neglected Tropical Diseases. As with all papers reviewed by the journal, your manuscript was reviewed by members of the editorial board and by several independent reviewers. The reviewers appreciated the attention to an important topic. Based on the reviews, we are likely to accept this manuscript for publication, providing that you modify the manuscript according to the review recommendations. 

Please pay close attention to the comments of the reviewers on the definition of persistent Brucellae and the experimental approaches required to demonstrate that indeed the populations surviving the action of antibiotics within macrophages belong to this category. 

Sincerely,

Esteban Chaves-Olarte

Guest Editor

Javier Pizarro-Cerda

Deputy Editor

Reviewer's Responses to Questions

**Key Review Criteria Required for Acceptance?**

**Methods**

-Are the objectives of the study clearly articulated with a clear testable hypothesis stated?

-Is the study design appropriate to address the stated objectives?

-Is the population clearly described and appropriate for the hypothesis being tested?

-Is the sample size sufficient to ensure adequate power to address the hypothesis being tested?

-Were correct statistical analysis used to support conclusions?

-Are there concerns about ethical or regulatory requirements being met?

Reviewer #1: Yes

Reviewer #2: The author should include a detailed section of statistical analysis section in Materials and Methods

**Results**

-Does the analysis presented match the analysis plan?

-Are the results clearly and completely presented?

-Are the figures (Tables, Images) of sufficient quality for clarity?

Reviewer #1: Yes

Reviewer #2: Yes

**Conclusions**

-Are the conclusions supported by the data presented?

-Are the limitations of analysis clearly described?

-Do the authors discuss how these data can be helpful to advance our understanding of the topic under study?

-Is public health relevance addressed?

Reviewer #1: Yes, mostly, except for the points raised in my comments to the authors.

Reviewer #2: Two limitations should be addressed. 1. Inhibition of bacterial replication in macrophages was only tested for 23 hours and no longer time points. 2. Antibiotic persisters were only tested with a single antibiotic instead of a combination.

**Editorial and Data Presentation Modifications?**

Reviewer #1: Minor comments:

1. Fig. 3A: how much of the intracellular bacterial population turn GFP+ upon aTc induction in cells not treated with antibiotics? This could be a useful control to include.

2. Fig. 3B is important as it is the only visualization provided for individual or small group of metabolically active bacteria that fit the definition of persisters. I suggest the authors magnify the insets of the split DsRed and GFP channels to better show these particular events in CIP-treated cells.

3. line 84: should be “regimen”.

4. lines 444-445: based on most of the available literature, Brucella replication does not usually start by 6 h pi, as the replicative vacuoles are not formed by then. The increase growth plotted by the authors in Fig. 4D likely reflects a lack of time points between 6 and 24 h that would otherwise show that the net increase in bacterial numbers does not happen until later, i.e. 8-12 h pi depending on the cellular models. While this comment may appear nitpicky, it is actually important with respect to my main point 2. 

5. Fig. S3C: This figure exemplifies the efforts taken by the authors to best control their microscopic analysis of persisters, but it would really help to add when the analysis shown in table S3C was performed. Was it done at 6 h pi, when one can assume that the wild type and VirB-deficient population still behave similarly?

Reviewer #2: (No Response)

**Summary and General Comments**

Reviewer #1: In this manuscript, Mode and colleagues have investigated whether the worldwide bacterial pathogen Brucella abortus generates persisters in response to clinically relevant antibiotic treatments during its intracellular cycle in macrophages as the potential cellular basis for relapses associated with antibiotherapies against human brucellosis. Using a combination of advanced, automated microscopy-based analyses of dual fluorescent reporters and enumeration of viable bacteria, the authors report that i) Brucella’s intracellular niche in RAW264.7 cells provides partial protection against antibiotics, which do not fully inhibit bacterial replication, ii) a subpopulation of intracellular bacteria remains metabolically active and cultivable, iii) Brucella form persister cells in culture and intracellularly that fully retain their infection capabilities. Based in these findings, they conclude that Brucella intracellular persisters may constitute a population that mediates relapses associated with brucellosis, despite antibiotic therapies, and suggest changes in clinical practices. 

This manuscript is clearly written and reports a well-designed and performed study that provides convincing evidence that a subpopulation of cultured or intracellular B. abortus display features consistent with a “persister” phenotype. As this study focuses on clinically relevant antibiotics and intracellular bacteria, it provides new information relevant to the treatment of brucellosis. I have a few concerns with the design of the microscopy-based detection of persisters and conclusions related to the intracellular source of persisters that need to be addressed, as detailed below.

1. By definition, persister cells should not grow or replicate intracellularly. It is therefore quite confusing that the fluorescence microscopy-based assay set up to identify intracellular persisters is based on the detection of foci of replicating bacteria inside cells during antibiotic treatments. These microcolonies clearly appear in the various micrographs provided as made of many bacteria (>50) so in no instance can they represent persister cells, which should show up as individual or small groups of bacteria (as shown for example in Fig. 3B). The authors define their assay as a readout of a population of intracellular bacteria that resist antibiotic treatment, which is technically correct, yet their replicative behaviour is inconsistent with the definition of persisters. These remaining so-called microcolonies may reflect instead phenotypic variations in the macrophage population, where individual cells may not take up antibiotics as well (or exclude them efficiently) and provide as a result a permissive niche for bacterial replication, regardless of persisters formation. As an example, Fig. 2E is presented to illustrate microcolonies in CIP-treated cells, but other cells in the field contain bacteria that display less replication and may represent true persisters. Yet, my understanding is that these were not counted in the analysis performed, or were they? This aspect of the study needs to be clarified.

2. A key part of the study is the demonstration of persisters formation in macrophages, which the authors associate with the protective environment of the intracellular niche and in particular the ER-derived replicative vacuole. However, the survival ratio remains relatively constant between 6, 24 and 48 h pi (Fig. 4D), while the ER-derived vacuole is not generated by 6 h pi in macrophages, even if bacteria start resuming growth (but not replication!) by then (Deghelt et al, 2014). This raises the question of the intracellular environment that promote persisters’ formation. The authors do not show whether persisters form in endosomal vacuoles (which are detectable throughout the whole cycle at frequencies consistent with persisters’ levels, even in the absence of antibiotics) or ER-derived vacuoles. In other words, does Brucella need to replicate in its ER-derived vacuole to form persisters? The authors make this claim but do not provide hard evidence for it. Is there an increased proportion of bacteria in endosomal (LAMP1+) vacuoles upon antibiotic treatment? This could be addressed by characterizing the intracellular niche of GFP+DsRed+ bacteria in antibiotic-treated cells.

Reviewer #2: General comment

This study aims to evaluate the antibiotic persistence of Brucella using a macrophage infection model. The authors use a Brucella strain carrying a dual reporter and evaluate bacterial persistence after antibiotic treatment by combining two techniques. (i) Microscopy of constitutive and induced reporters, and (ii) CFU determination. They conclude that antibiotic treatment fails to prevent the growth B. abortus in cellulo. They attribute the persistence of Brucella to the protective nature of the intracellular niche and the formation of intracellular persisters.

The authors explore a topic (barely explored experimentally) quite relevant to improving brucellosis treatment. The assays are well-validated, the results well explained, and the conclusions supported by their results.

Minor concerns

Inhibition of bacterial replication in macrophages was only tested after 23 hours (Figure 2). Is there a reason for using this time point? 

Was bacterial replication evaluated for more extended periods? This could be relevant considering that Brucella treatments require multiple doses.

Why were antibiotic persisters tested with a single antibiotic instead of a combination since it was shown that single treatments do not work? 

Add a detailed statistical analysis section to the materials and methods section.

PLOS authors have the option to publish the peer review history of their article (what does this mean?). If published, this will include your full peer review and any attached files.

Reviewer #1: No

Reviewer #2: Yes: ELIAS BARQUERO-CALVO

Figure Files:

Data Requirements:

Reproducibility:

References

---

## [Decision Letter · Decision Letter 1]

5 Jul 2022

Dear Dr. Dehio,

We are pleased to inform you that your manuscript 'Antibiotic persistence of intracellular Brucella abortus' has been provisionally accepted for publication in PLOS Neglected Tropical Diseases.

Best regards,

Esteban Chaves-Olarte

Guest Editor

Javier Pizarro-Cerda

Deputy Editor

Reviewer's Responses to Questions

**Key Review Criteria Required for Acceptance?**

**Methods**

-Are the objectives of the study clearly articulated with a clear testable hypothesis stated?

-Is the study design appropriate to address the stated objectives?

-Is the population clearly described and appropriate for the hypothesis being tested?

-Is the sample size sufficient to ensure adequate power to address the hypothesis being tested?

-Were correct statistical analysis used to support conclusions?

-Are there concerns about ethical or regulatory requirements being met?

Reviewer #1: All criteria are met

Reviewer #2: (No Response)

**Results**

-Does the analysis presented match the analysis plan?

-Are the results clearly and completely presented?

-Are the figures (Tables, Images) of sufficient quality for clarity?

Reviewer #1: Yes

Reviewer #2: (No Response)

**Conclusions**

-Are the conclusions supported by the data presented?

-Are the limitations of analysis clearly described?

-Do the authors discuss how these data can be helpful to advance our understanding of the topic under study?

-Is public health relevance addressed?

Reviewer #1: Yes

Reviewer #2: (No Response)

**Editorial and Data Presentation Modifications?**

Reviewer #1: (No Response)

Reviewer #2: (No Response)

**Summary and General Comments**

Reviewer #1: In this revised version of their manuscript, the authors have thoroughly addressed original concerns to this reviewers satisfaction. I have no further comments.

Reviewer #2: The authors have addressed the queries satisfactorily by giving explanations to some of the questions and by adding additional data. The manuscript is acceptable for publishing.

PLOS authors have the option to publish the peer review history of their article (what does this mean?). If published, this will include your full peer review and any attached files.

Reviewer #1: No

Reviewer #2: **Yes: **Elías Barquero-Calvo

---

## [Editor Report · Acceptance letter]

22 Jul 2022

Dear Prof. Dehio,

We are delighted to inform you that your manuscript, "Antibiotic persistence of intracellular *Brucella abortus*," has been formally accepted for publication in PLOS Neglected Tropical Diseases.

Best regards,

Shaden Kamhawi

co-Editor-in-Chief

Paul Brindley

co-Editor-in-Chief
